# Direct Electrochemical Detection of Glutamate, Acetylcholine, Choline, and Adenosine Using Non-Enzymatic Electrodes

**DOI:** 10.3390/s19030447

**Published:** 2019-01-22

**Authors:** Arash Shadlaghani, Mahsa Farzaneh, Dacen Kinser, Russell C. Reid

**Affiliations:** Department of Mechanical and Energy Engineering, University of North Texas, Denton, TX 76209, USA; arashshadlaghani@my.unt.edu (A.S.); mahsa.farzaneh@unt.edu (M.F.); dacenkinser@my.unt.edu (D.K.)

**Keywords:** non-electroactive neurotransmitter, electrochemical biosensor, non-enzymatic sensing, nanobiological materials

## Abstract

Non-electroactive neurotransmitters such as glutamate, acetylcholine, choline, and adenosine play a critical role in proper activity of living organisms, particularly in the nervous system. While enzyme-based sensing of this type of neurotransmitter has been a research interest for years, non-enzymatic approaches are gaining more attention because of their stability and low cost. Accordingly, this focused review aims to give a summary of the state of the art of non-enzymatic electrochemical sensors used for detection of neurotransmitter that lack an electrochemically active component. In place of using enzymes, transition metal materials such as those based on nickel show an acceptable level of catalytic activity for neurotransmitter sensing. They benefit from fast electron transport properties and high surface energy and their catalytic activity can be much improved if their surface is modified with nanomaterials such as carbon nanotubes and platinum nanoparticles. However, a general comparison reveals that the performance of non-enzymatic biosensors is still lower than those that use enzyme-based methods. Nevertheless, their excellent stability demonstrates that non-enzymatic neurotransmitter sensors warrant additional research in order to advance them toward becoming an acceptable replacement for the more expensive enzyme-based sensors.

## 1. Introduction

Neurotransmitters play a crucial role in daily behavioral, physiological, and neurological functions such as learning, sleeping, consciousness, and heart regulation [1]. Consequently, a problem in the metabolism of these chemicals or their abnormal level can be associated with various mental and physical disorders including Parkinson’s and Alzheimer’s diseases, congestive heart failure, epilepsy, and so on. Clearly, neurotransmitters are clinically relevant biomarkers, so their accurate and real-time monitoring in physiological structures is essential for effective diagnosis and therapeutic interventions [2,3,4]. Sensing and measuring of neurotransmitters is also necessary in many other applications such as food safety, drug discovery, and environmental monitoring.

Electrochemical techniques are highly advantageous in real-time quantitative analysis of neurotransmitters because electrochemical methods are generally a faster and more user-friendly analysis tool than classical techniques [5]. Enzyme-based electrochemical sensing is a common neurotransmitter detection approach because of the non-electroactive nature of some neurotransmitters. However, enzyme-based biosensors can be expensive due to the high cost of enzymes and can also suffer from the lack of long-term stability caused by enzyme denaturation. Thus, simpler methods such as non-enzymatic biosensors have received more attention during recent years, which is the scope of present review.

The main contribution of this review compared to previous neurotransmitter sensor reviews is that it investigates the merits and challenges of non-enzymatic biosensors, specifically for detecting acetylcholine, glutamate, choline, and adenosine triphosphate (ATP) [6]. In contrast to neuro-transmitters such as dopamine, the aforementioned neurotransmitters cannot easily be directly monitored using electrochemical methods at moderate applied potentials and neutral pH. Rather, they must first be converted to, or must produce, a more easily electrolyzed component in order to be detected. Due to the difficulty in electrochemically detecting these neurotransmitters in their native media and at a reasonable potential, the term “non-electroactive” is commonly applied to these neurotransmitters [7,8] and they will be referred to as such here. A recent review by Chandra et al. [9] dedicated a small portion to non-enzymatic neurotransmitter biosensors but only discussed detection of electroactive neurotransmitters such as dopamine whereas the present review focuses on non-electroactive neurotransmitter detection. In addition, the role of electrode materials in sensing non-electroactive neurotransmitters is discussed. It is important to better understand the performance of non-enzymatic sensors since they are a potentially inexpensive, stable option for many applications, particularly clinical applications. Increased understanding regarding non-enzymatic sensors including material biocompatibility and their sensitivity and bioselectivity will progress this type of sensor toward further applications such as in vivo capability, which remains a notable obstacle. Hereby, this paper highlights the main progress in non-enzymatic sensing of these select neurotransmitters while mentioning the challenges that need to be addressed.

This focused review is divided into five sections including the introduction. Section 2 gives a brief overview of electrochemical neurotransmitter detection and existing measurement challenges either enzymatically or non-enzymatically. Section 3 characterizes materials used in non-enzymatic detection of non-electroactive neurotransmitters. Section 4 compares the performance of enzymatic vs. non-enzymatic non-electroactive neurotransmitter detection. Finally, Section 5 looks at opportunities and challenges in terms of in vivo applicability. 

## 2. Neurotransmitter Characteristics

Neurotransmitters are endogenous chemical substances that play a critical role in regulating neuron transmembrane potential. They are released at the end of a nerve fiber in response to the arrival of a nerve impulse and allow transferring the impulse from one nerve fiber to the next nerve fiber, muscle fiber, or other structures throughout the nervous system [10]. Neurotransmitters can be classified according to their molecular structure, mode of action (either direct or as a neuromodulator) and physiological function (either excitatory or inhibitory). Neurotransmitters can also be classified as either electroactive or non-electroactive depending on their response to an applied electric field, which determines whether or not they can be directly detected by electrochemical methods. Glutamate, choline, acetylcholine, and ATP are examples of non-electroactive neurotransmitters, which have been of interest to many researchers because of their vital function in the brain. Glutamate is a key excitatory neurotransmitter in the central nervous system because of its involvement in many aspects of normal brain functioning. Increased concentration of extracellular glutamate in the synaptic cleft can lead to excessive calcium entry, which is the main cause of neuronal injury and death by stroke, cerebral hypoxia/ischemia, etc. [11]. Acetylcholine is another neurotransmitter that intercedes the chemical transmission of neuronal signals at synapses in the central and peripheral nervous system in order to regulate physiological levels of neurotransmitters and bursting mode of neuronal firing [12]. Choline has essential roles in brain development and memory function of infants and adults and it is important in the synthesis of some essential phospholipids that provide structure to cell membranes and facilitate transmembrane signaling [13]. ATP that accumulates in the extracellular space at sites of damaged tissue is efficacious in the cardiovascular, nervous, gastrointestinal and immune system [14]. Given neurotransmitters’ crucial role, monitoring their concentration is very important in diagnosing the nature of problems caused by their variations. 

### 2.1. Neurotransmitter Detection

Neurotransmitters can be measured by different techniques including chromatographic, spectrophotometric, fluorimetric, laser, liquid chromatography, chemical assays, etc. Although these methods have sufficient strengths tailored for certain target applications, each also has shortcomings such as being labor-intensive, requiring expert handling and/or large equipment, and being time consuming [15] that reduce their widespread usage. Significant recent attention has been devoted to the electrochemical approach due to its practical advantages which include high sensitivity and selectivity, rapid response, low cost, operational simplicity, easy miniaturization, etc. [5] all of which make it an excellent choice for continuously monitoring in real-time the dynamic behaviors of the various neurotransmitters.

One main challenge of electrochemical biosensors is creating a surface architecture that leads to a high electronic response (sensitivity). To overcome this problem, shrinking the dimensions of electrochemical sensor elements to sizes which can increase the signal-to-noise ratio have been increasingly of interest [16]. There are many recent reviews (only a few are mentioned here) focusing on remarkable advances in the development of novel ultrasensitive electrochemical assays, including enzyme-based or enzyme-less sensors utilizing nanomaterials and nanostructures [17,18,19,20]. There are also very good reviews specifically focusing on neurotransmitter detection [9,21]. However, non-enzymatic detection of non-electroactive neurotransmitters was outside the scope of previous reviews. 

### 2.2. Challenges of Measuring Neurotransmitters

Biosensors must produce sensitive and selective analytical signals, which are pertinent to analyte concentration. Electrochemical biosensors require a transduction mechanism (i.e., a redox reaction) to convert physiochemical changes between a target and the corresponding biorecognition element(s) into quantifiable electronic signals. The main challenges for in vivo neurotransmitter monitoring are fast response time related to rapid neurotransmitter release and clearing from the extracellular space, low concentrations [8], large background signal and noise, and device fouling and degradation over time. In vitro sensors must address these challenges as well in order to demonstrate eventual clinical applicability. Achievements in nanotechnology and nanoscience demonstrate that nanobiological materials including metal nanoparticles, quantum dots (QDs), and carbon nanotubes (CNTs) have attracted much attention because of their unique material and size-dependent physiochemical properties [22]. For example, metal and carbon-based nanoparticles and fibers have great potential for improving detection because they produce a synergic effect among catalytic activity, conductivity, and biocompatibility to speed up signal transduction [23,24]. Moreover, these materials can amplify biorecognition events with specifically designed signal tags, leading to highly sensitive biosensing [25]. Despite metallic nanoparticles’ improved electrochemical, mechanical, optical and magnetic biosensor properties, their development towards clinical applications is still a challenge since their interaction with the immune system is somewhat unknown [26]. Considering that biological molecules possess special structures and functions, the mechanical and electrical stability as well as the biocompatibility of nanomaterials in specific physiological environments are critical [27]. 

### 2.3. Enzymatic vs. Non-Enzymatic Electrochemical Neurotransmitter Biosensors

In order to detect non-electroactive neurotransmitters, enzymatic sensors generally utilize an enzyme agent to enhance the electrocatalytic activity of the corresponding neurotransmitter on the electrode surface by generating more easily detectable compounds (e.g., hydrogen peroxide). However, enzyme cost is often an issue in the fabrication of this kind of neurotransmitter sensor [28]. Other shortcomings that need to be overcome are complicated enzyme purification procedures, lack of long-term stability due to enzyme denaturation, reproducibility, reusability, and low sensitivity owing to indirect electron transfer [29]. In contrast, non-enzymatic sensors can be made with materials having excellent catalytic activity and therefore don’t necessarily require enzyme immobilization onto the electrode surface.

For example, using highly conductive metallic nanomaterials with intrinsic catalytic activity can facilitate transfer of electrons between analyte and electrode [30]. Also, the large surface area and high mechanical strength of these nanomaterials are beneficial in electrochemical sensors. Drawbacks include surface contamination/fouling, optimal operation in non-physiological conditions, and sometimes complicated preparation procedures [9]. A schematic illustrating enzymatic and non-enzymatic neurotransmitter electrochemical detection is shown in Figure 1.

## 3. Materials for Non-electroactive Neurotransmitter Detection

This section highlights the most common materials used for non-enzymatic electrochemical detection of glutamate, acetylcholine, choline, and ATP. Although this section does not discuss all the relevant literature examples pertaining to this topic, Table 1 contains the majority of recent reported non-enzymatic electrochemical biosensors for these neurotransmitters.

### 3.1. Electrochemical Reactions on Non-Enzymatic Electrodes

Non-enzymatic electroanalytic neurotransmitter sensing is generally accomplished through oxidation of the neurotransmitter on a transition metal in alkaline media. Equations (1) and (2) show the basic reaction equations using nickel as an example transition metal because it is the most common material used. In these equations, the Ni(II)/Ni(III) redox couple is employed and Ni(OH)_2_ is oxidized to NiOOH, and then NiOOH is reduced to Ni(OH)_2_ by neurotransmitter oxidation. Equation (2) is actually a multi-step reaction involving various organic intermediates each reacting with the transition metal hydroxide [23]:(1)$$\mathrm{Ni}{\left(\mathrm{OH}\right)}_{2}+{\mathrm{OH}}^{-}\to \mathrm{NiOOH}+H_{2}O+e^{-}$$
(2)$$\mathrm{NiOOH}+\mathrm{Neurotransmitter}\to \mathrm{Ni}{\left(\mathrm{OH}\right)}_{2}+\mathrm{product}$$

Enzymatic non-electroactive neurotransmitter detection utilizes enzymes immobilized in a bioselective membrane. The enzyme catalyzes the oxidation of neurotransmitters, which results in the formation of an electrochemically active component, normally hydrogen peroxide. Therefore, a commonality between enzymatic and non-enzymatic detection of non-electroactive neurotransmitters is that both approaches involve indirect monitoring of the target analyte.

Figure 2 compares the detection of two distinct examples of non-enzymatic glutamate sensors, where the working electrodes was a nickel nanowire array electrode, NiNAE, (Figure 2a) and a glassy carbon electrode, GCE, modified with NiO nanoparticles (Figure 2b) [31]. Both plots in Figure 2 show an increased anodic peak current upon the addition of glutamate in a NaOH solution, exhibiting electrocatalytic activity towards the oxidation of glutamate, non-enzymatically. The addition of glutamate also shifts the anodic peak to the higher potential, which might be pertinent to glutamate diffusion limitation at the electrode [28]. It is worthy to note that the current density is higher for the NiNAE indicative of its active higher surface area compared to the NiO nanoparticles on GCE.

Cyclic voltammogram responses of two modified electrodes to the presence of ACh are shown in Figure 3. The Ni(II)/Ni(III) redox couple forming in the alkaline medium [32] is clearly shown in the oxidation and reduction peaks at roughly 0.4 V and 0.6 V vs. Ag/AgCl, respectively. With the addition of acetylcholine, reduction peak currents decreased while oxidation peaks increased indicating that a portion of the NiOOH was reduced through acetylcholine oxidation. Note that Figure 3a has been rotated to be consistent with the convention of the other presented plots.

While not shown in Figure 3, using carbon dots, CDs, and ordered mesoporous carbon, OMC, considerably improved electrocatalytic activity. Wang et al. [23] also showed that the ratio of carbon nanomaterial to layered double hydroxides (LDH) is critical and needs to be optimized. Based on their results, the oxidation peak current of acetylcholine rose as the CD/LDH mass ratio increased from 0 to 0.025 because of carbon dot adsorption on the modified electrode surface. In contrast, a higher ratio increased the water solubility of the NiAl-LDH/CD composites, leading to a reduction in composite deposited on the working electrode surface, which consequently decreased oxidation peak current. Note that the cyclic voltammograms in the figures were performed at non-physiological analyte concentrations for redox characterization because low in vivo neurotransmitter concentration makes it difficult to visualize oxidation/reduction peaks. The issue of in vivo sensor applicability is discussed in more detail in Section 5.

### 3.2. Commonly Used Materials

Nickel is one of the most promising electrode materials for non-enzymatic, nonelectroactive neurotransmitter sensors because of its electrochemical redox reactivity, fast electron transport properties, natural abundance, environmental friendliness, high surface energy, and low cost. Nickel thin films can be deposited using conventional methods and nickel nanoparticles and nanowires can be synthesized with minimal chemicals and equipment (see [28,31] for examples). When immobilized as hydroxides, nanoparticles, thin films, etc. at an electrode surface, the Ni(II)/Ni(III) redox chemistry described in Equations (1) and (2) can be coupled with neurotransmitter oxidation.

Ni-based sensors constitute the majority of studies relevant to this review. For example, layered double hydroxides (LDHs) having intercalated nickel and aluminum hydroxide layers, have been used for non-enzymatic electrochemical neurotransmitter biosensors [23,32]. Although they suffer from poor electroconductivity, LDHs benefit from a permanent positively charged layer, large surface area, good adsorption ability and anion exchange properties [23] while degradation and poisoning of the electrode can occur under applied potential. Figure 4a shows an SEM image of a nickel, aluminum-based LDH (NiAl-LDH) used for acetylcholine detection [23]. It can be observed that the convoluted surface structure of the LDH provides a larger active surface area. To further improve nickel-based LDH catalytic activity and surface area for non-enzymatic acetylcholine sensing, Wang et al., Ju et al. [23,32] employed carbon dots (CDs) and ordered mesoporous carbon (OMC), respectively, for modifying glassy carbon working electrodes. Because of higher specific surface area and large pore volume, the presence of OMC augmented charge transfer of Ni–Al LDHs, leading to at least three times higher current than using CDs.

Nickel oxide (NiO), having acceptable durability and stability, is another attractive material in non-enzymatic neurotransmitter sensors [41,49,58]. NiO is a hole-type semiconductor that facilitates electron transfer tremendously and has non-toxic properties. Jamal et al. [31] drop-casted a mixture of NiO nanoparticles, chitosan and Nafion on the surface of a GCE to study glutamate detection.

In a study on acetylcholine detection [41], a non-enzymatic sensor was fabricated using high surface area “lichen-like” NiO, which produced a higher acetylcholine oxidative current compared to Ni micro or nanoparticles. Similarly, flower-like NiO was used by Sattarahmady et al. [49] for choline detection.

Different forms of nickel, including wire [44], thin films [43], nanoshells [40], and nanowires [42] have also been utilized. Lin and Chou [51] reported an acetylcholine sensor consisting of electrodeposited Ni film on carbon rods. Another acetylcholine sensor revealed the sensing capability of hollow nickel microspheres mixed with carbon microparticles, both of which were immobilized in a Nafion matrix [40]. 

A glutamate sensor employing an electrodeposited vertically aligned nickel nanowire array Figure 4b was reported by Jamal et al. [28]. The authors demonstrated that adding platinum particles to the nickel nanowire array increased sensitivity to acetylcholine. Besides being used its more conventional metallic forms, nickel-based detection has even been accomplished with nickel as the center atom in a metalloporphyrin. Carballo et al. [50] demonstrated that direct amperometric acetylcholine and choline detection is possible using poly[Ni(II)protoporphyrin IX] (pNiPP). The sensing mechanism in that case was not Faradaic in nature, but rather involved the perturbation of steady-state ionic flow through the pNiPP polymer when the larger acetylcholine and choline molecules interacted with the membrane. This resulted in charge redistribution and acetylcholine or choline concentration-dependent current responses. One drawback to this approach was the response was highly dependent on Cl- concentration and could be almost completely inhibited with 0.1 M KCl.

In addition to nickel-based materials, other materials have also been used for nonelectroactive neurotransmitter detection. Another transition metal, copper, can be used for non-enzymatic neurotransmitter sensing as demonstrated by Heli et al. [36] who mixed copper particles with carbon powder to make modified carbon paste electrodes for acetylcholine sensing. Oxidation of acetylcholine on the copper-carbon paste electrode was likely accomplished through the Cu(II)/Cu(III) couple similar to the Ni-based examples. Besides nickel and copper, silver has also been utilized for non-enzymatic neurotransmitter sensing. A non-enzymatic aptasensor utilizing silver nanoparticles (AgNPs) for amplification was developed to detect adenosine triphosphate (ATP) [52]. Aptameric sensors are non-enzymatic detectors using binding oligonucleotides that are highly specific to target analytes such as ATP. Protein binding produces conformational changes or desorption which, in the case of [52], resulted in a change to the available AgNP electroactive surface area. Finally, a potentiometric choline sensor was demonstrated [45] wherein a synthetic cavitand (container-shaped molecule) receptor was co-immobilized with single-walled CNTs (SWCNTs). Choline selectively bound to the cavitand to produce an electrochemical boundary potential that was dependent on choline concentration.

Table 1 summarizes the key performance parameters of non-enzymatic detection of non-electroactive neurotransmitters. It should be mentioned that most electrochemical studies were performed in a three-electrode configuration containing NaOH analyte. The basic analyte solution is necessary to supply the OH- used for Ni(OH)_2_ oxidation as shown in Equation (1). Although NaOH benefits from high ionic conductivity due to the small size and high mobility of the OH- anion, relying on NaOH precludes sensor use for in vivo applications.

## 4. Enzymatic vs. Non-Enzymatic Neurotransmitter Sensors

On a basic level, enzymatic and non-enzymatic neurotransmitter detection differs in catalyst identity: non-enzymatic sensing operates through metallic catalysts whereas enzymatic detection uses enzyme catalysts. Both methods fundamentally involve metal atom redox chemistry although for non-enzymatic sensing this occurs at the metal surface and for enzymatic catalysis the metal atoms are often buried within the protein structure, making it challenging electron transfer to occur. This leads to a theoretical advantage for non-enzymatic sensors because analyte detection can occur more readily, but it also enables competing surface reactions from interferents and this can negatively impact sensitivity and limit of detection (LOD). Enzymes, however, are generally not as stable outside of their native membrane-bound and/or biological environment. This is why enzyme immobilization is critical for enzymatic sensing. 

The right-hand columns of Table 1 contain representative performance data for enzymatic electrochemical sensors of glutamate, acetylcholine, choline, and adenosine. These values represent excellent sensitivity and LOD values for previously reported enzymatic sensors. A comparison of enzymatic and non-enzymatic neurotransmitter sensors in Table 1 indicates that enzymatic neurotransmitter sensors currently outperform non-enzymatic sensors, although the highest enzymatic performance metrics for each analyte all come from different examples. It would be highly unlikely to find an enzymatic sensor with superior sensitivity, LOD, and stability. Nevertheless, some useful conclusions can be drawn by comparing magnitudes for sensitivity and LOD. For example, enzymatic glutamate and choline LOD appear to have a clear advantage over non-enzymatic choline LOD because they are at least three orders of magnitude lower. On the other hand, non-enzymatic acetylcholine sensitivity is clearly higher than the enzymatic sensors. Also, the non-enzymatic adenosine aptasensors have a significantly lower LOD compared to what is available with enzymatic sensing.

The sensitivity and LOD advantages of enzymatic vs. non-enzymatic sensors can be partially explained by the comparative number, and therefore scientific progress, of enzymatic vs. non-enzymatic studies that have been performed. The number of these studies is partially illustrated with recent sensor reviews/summaries for glutamate [59], choline [60], and adenosine [61]. Enzymatic sensors may also outperform non-enzymatic ones due to higher signal-to-noise ratios stemming from enzyme-substrate selectivity and lower operating potential than typical non-enzymatic sensors. For example, acetylcholine is oxidized through the Ni(II)/Ni(III) couple at a higher voltage for a non-enzymatic biosensor, 0.46 V vs. Ag/AgCl [42], than for an enzymatic one, 0.19 V vs Ag/AgCl [62]. Lower operating voltage increases the usability of enzymatic biosensors for in vivo applications because of lower oxidation currents from interfering biomolecules.

Common interferents co-existing with non-electroactive neurotransmitters include ascorbic acid, uric acid, glucose, and dopamine. It is generally accepted that enzymatic sensors are more selective than non-enzymatic but practical selectivity comparisons are quite difficult given the study-to-study differences in experimental conditions. For example, Wang et al. [23] showed that carbon dots (CDs) increased selectivity of their acetylcholine sensor since the CDs with negative surface charge could attract the acetylcholine with positive charge. The interference rates of dopamine, ascorbic acid, and norepinephrine were 12%, 9.7% and 4.3%, respectively. The amount of ascorbic acid was about 250 µM in KOH solution, which contained 50 µM acetylcholine. However, Bolat et al. [62] indicated 3.7% reduction in the response of an enzyme-modified glassy carbon electrode to 100 µM ascorbic acid in a solution containing 1 mM acetylcholine. So, the enzymatic sensor demonstrated a lower interference rate due to ascorbic acid but the molar ratio of ascorbic acid to acetylcholine was also much lower: 0.1 vs. 5. In other examples, the comparison is more direct. For instance, Jamal et al. [31] investigated the selectivity of their glutamate sensor to ascorbic acid and uric acid. Amperometric responses illustrated that 100 µM ascorbic acid and uric acid yielded a decreased current response of 20% and 4%, respectively, compared to 1 mM glutamate [28]. Batra et al. [34] showed that the addition of 1 mM ascorbic acid into a solution containing ≤400 µM glutamate reduced the selectivity of a GluOx/PPyNPs/PANI modified Au electrode about 3%. Therefore, the presence of enzyme increased the selectivity of the biosensor. 

Yabuki [63] showed that immobilizing enzymes in membrane materials is effective in the stability of an enzyme-based sensor. While non-enzymatic sensors are not subject to this requirement, they can be susceptible to biofouling. Like comparing selectivity, comparing long-term stability of enzymatic and non-enzymatic sensors is also challenging due to a wide variety of experimental parameters such as storing conditions, analyte concentration, continuous or intermittent contact of the biosensor with the analyte solution, pH, temperature, etc. All these factors can affect stability and without clear guidelines regarding how to test and report long-term stability, truly quantitative comparisons between studies are impossible. There are, however, notable stability examples from Table 1 such as the non-enzymatic glutamate sensor by Jamal et al. [28] who tested their electrodes five times every two days for four weeks after which time the sensor still retained 96% of its initial activity. The same group performed a similar stability test on another non-enzymatic glutamate sensor, this time testing the electrodes every day for 15 days with the electrode stored at room temperature between experiments; 99.2% of the sensor’s initial activity was maintained. These examples are exceptional given the rigorous level of testing compared to typical stability testing involving storage in a cold, dry environment and a single subsequent experiment at the end of the waiting period. This is the type of long-term stability testing typically performed on enzymatic neurotransmitter sensors. As an example, Soldatkina et al. [64] studied the stability of their enzymatic glutamate sensor by daily measuring nine responses to adding 1 mM glutamate during 4 days. During the measurements, the biosensor remained in the buffer at continuous stirring and after the measurements, it was kept in a refrigerator. 

## 5. In Vivo Applicability

With the exception of the aptasensors for adenosine detection, most of the reported non-enzymatic sensors cannot currently be used for in vivo applications because of the required alkaline pH and relatively high operational voltage. The catalytic activity of the metallic nanomaterials depends on the concentrated OH- anions meaning that their catalytic efficiency toward oxidation of non-electroactive neurotransmitters will be insignificant in either neutral or acidic conditions. Additional research is needed to modify Ni-based materials to operate at lower pH. To evaluate the effect of pH on ATP sensing, Mashhadizadeh et al. [52] investigated current response of a silver nanoparticles decorated graphene oxide electrode. Based on the structure of the graphene oxide, there was an optimum pH for peak oxidation of the working electrode. Although studies such as this demonstrate the benefit of higher pH for the nickel redox reaction, it also shows there is room for optimization. This is encouraging because it indicates that there may be a material with a broad operating pH that can enable non-electroactive neurotransmitter catalytic activity at neutral pH. However, enzymatic sensing has already shown that it can be applied in neutral solution. For example, Govindarajana et al. [65] fabricated an enzyme-based glutamate sensor, which was independent from pH value and could be used in a neutral pH ≈ 7 solution. 

Moreover, neurotransmitters are also highly diffusive, and their release levels are extremely low and yet the concentration of neurotransmitters used in the reviewed papers was much higher than in bodily fluids. For example, the mean blood glutamate concentration for humans is about 60 µM which can be ultimately increased to 80 µM by traumatic brain injury [66]. However, 1 mM was the minimum glutamate concentration that was investigated by Jamal et al. [31]. The blood acetylcholine concentration of humans is about 1.26 nM but the minimum concentration that Wang [23] studied was 5 µM. Therefore, more attempts are required to make a really practical non-enzymatic biosensor to non-enzymatically detect physiologically relevant concentrations of these neurotransmitters. It should be noted that enzyme-based sensors are already capable of sensing appropriate levels of such neurotransmitters [34]. As examples, Wahono et al. [67] fabricated a needle type microsensor with several permselective membranes which could detect lower concentration of glutamate (around 20 µM). Wei et al. [68] designed a novel implantable microelectrode array probe for producing simultaneous measurements of glutamate and electrophysiological recordings. Their sensor could detect ≤10 µM of glutamate.

Another important challenge related to in vivo applicability of non-enzymatic neurotransmitter sensors is electrochemically active interferents, which generate electrochemical signals at the same redox potential as the target neurotransmitter. For example, glucose is a major oxidizable substance and it exists in many of the same biological fluids as target neurotransmitters. Like the neurotransmitters discussed in this review, it can be oxidized on nickel-based materials in an alkaline solution [69] and at a potential that is very close to the operating voltages of Ni-based non-enzymatic sensors. Therefore, additional research is needed to identify non-enzymatic redox couples catalyzing non-electroactive neurotransmitter oxidation at lower potentials.

## 6. Conclusions

Recent progress in the field of non-electroactive neurotransmitter sensing, particularly the non-enzymatic approach, have been discussed in this short review. In order to detect non-electroactive neurotransmitters such as glutamate, acetylcholine, choline, and adenosine, which are vital chemical substances in the nervous system, electrochemical biosensors have a promising accuracy. Although enzyme-based sensors have been widely used in neurotransmitter sensing, non-enzymatic approaches are also gaining much interest because of their simplicity, low-cost, and stability.

Given their larger surface area, LDH are a propitious structure for non-enzymatic neurotransmitter detection due to its micro/nanodimensional structure and anion exchange properties for adsorption of biomolecules. Higher conductivity of some metal oxides such as nickel oxide are also beneficial to enhance the catalytic activity of electrodes; however, their electrical conductivity is not as amenable for facile electron transfer. Carbon nanomaterials are highly recommended to overcome this shortcoming. In this context, Niu et al. [70] have stated that low cost reproducibility of these nanosized materials is a big challenge for large-scale practical applications.

The information presented in this article shows that non-enzymatic sensors have the potential of being an acceptable approach to detect non-electroactive neurotransmitters; however, much more research is needed to improve their electrochemical performance, in vivo applicability, and long-term stability.

## Figures and Tables

**Figure 1 sensors-19-00447-f001:**
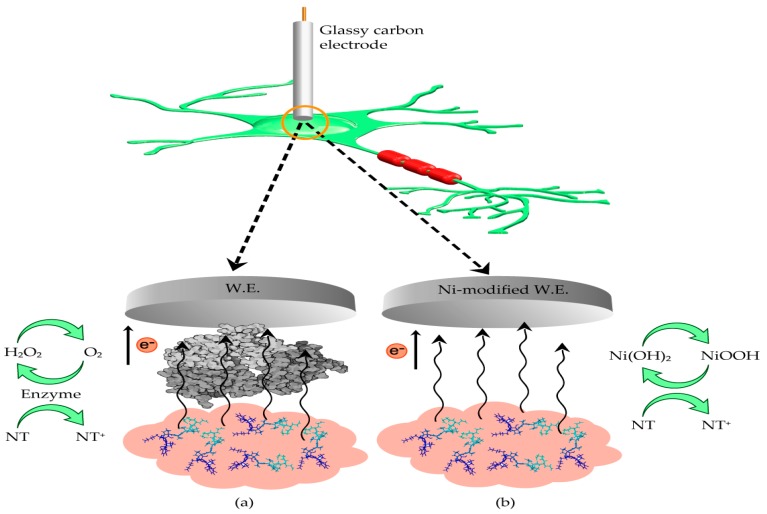
Schematic of neurotransmitter detection methods at working electrode (WE) surface: (**a**) enzymatic sensing; (**b**) Non-enzymatic sensing using a nickel-based material as an example.

**Figure 2 sensors-19-00447-f002:**
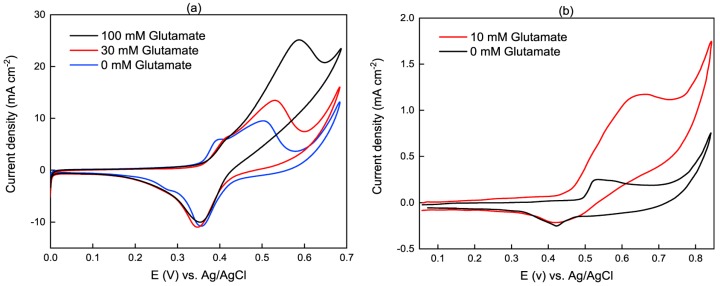
Example cyclic voltammetry of non-enzymatic glutamate sensors. (**a**) NiNAE electrode in 1 M NaOH with a scan rate of 0.04 V/s; (**b**) NiO/GCE electrode in 0.1 M NaOH with the scan rate of 0.05 V/s. Reproduced with permission from [28] and [31] copyright [2013], [2018], Elsevier and Springer.

**Figure 3 sensors-19-00447-f003:**
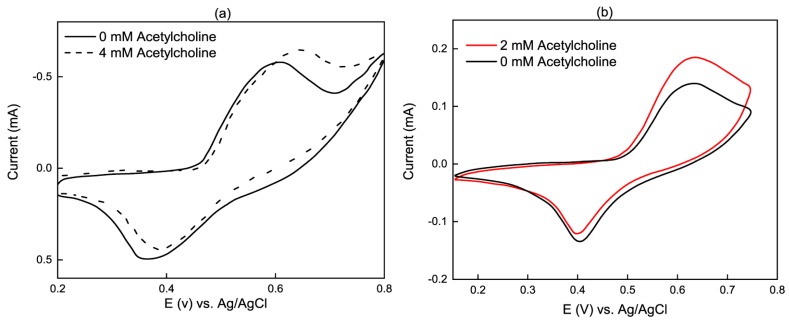
Cyclic voltammetry of acetylcholine sensors at 0.1 V/s. (**a**) Ni–Al LDHs/OMC/GC in 0.1 M NaOH; (**b**) NiAl-LDH/CD/GCE in 0.1 M KOH. Reproduced with permission from [32] and [23] copyright [2012], [2016], Elsevier.

**Figure 4 sensors-19-00447-f004:**
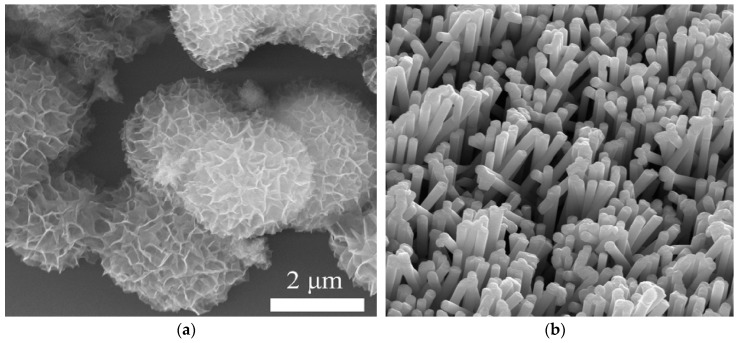
SEM image of NiAl-LDHs (**a**) [23] and Ni nanowire array electrode (**b**) [28]. Reproduced with permission from [23] and [28] copyright 2016, 2013 Elsevier.

**Table 1 sensors-19-00447-t001:** Materials and performance of non-enzymatic electrochemical neurotransmitter biosensors

Ref.	Electrode [Primary Electrolyte]	Sensitivity (µA mM^−1^ cm^−2^)	LOD (µM)	Linear Range (mM)	Stability (% Activity)	Enzymatic Sensors		
						Sensitivity (µA mM^−1^ cm^−2^)	LOD (µM)	Stability (%Activity)
**Glutamate**								
[28]	Pt-NiNAE [1 M NaOH]	96	83	0.5–8.0	4 weeks (96%)	433 [33]	1 × 10^−4^ [34]	20 weeks [35]
[31]	NiO/GCE [0.1 M NaOH]	11	272	1.0–8.0	2 weeks (99%)			
**Acetylcholine**								
[36]	Cu/CPE [0.1 M NaOH]	-	39	0.12–2.68	-	2.19 [37]	0.004 [38]	2 weeks [39]
[40]	ns-Ni-CC [0.1 M NaOH]	48.58	0.049	0.0002–0.828	5 weeks			
[32]	NiAl-LDHs/OMC/GCE [0.1 M NaOH]	-	0.042	0.002–4.92	3 weeks (93%)			
[41]	NiO/CPE [0.1 M NaOH]	392.4	26.7	0.25–5.88	-			
[23]	NiAl-LDH/CD/GCE [0.1 M KOH]	133.2	1.7	0.005–6.88	8 weeks (95%)			
[42]	NiNAE [0.1 KOH]	~1400	0.84	-	-			
[43]	Ni/Pt-graphite [0.2 NaOH]	-	~0.1	0–4 × 10^−4^	-			
[44]	Ni wire [0.2 NaOH]	~9.5	-	0–1.7	-			
**Choline**								
[45]	Cavitand+SWCNT/GCE	-	0.39	0.010–10.0	12 weeks	495 [46]	1 × 10^−4^ [47]	13 weeks (95%) [48]
[49]	Flower-like NiO [0.1 NaOH]	60.5	25.4	0.25–6.98	4 weeks			
[50]	pNiPP [0.3 M NaCl]	~6	18	-	-			
[51]	Ni [0.2 M NaOH]	~97	~134					
**Adenosine**								
[52]	Aptamer-AgNPs/graphene oxide	-	0.005	1 × 10^−5^–85 × 10^−5^	-	250 [53]	0.010 [54]	7 weeks (57%) [55]
[56]	Aptamer/QDs [0.2 M acetate]	~1.8	0.03	1 × 10^−8^–0.1	-			
[57]	Aptamer/MB/Au [0.3 M NaCl]	-	0.0072	0–5 × 10^−5^	-			

Pt-NiNAE = platinum-modified nickel nanowire array electrode, ns-Ni-CC = nickel nanoshell-carbon microparticles, LDH = layered double hydroxide, pNiPP = Poly[Ni(II)Protoporphyrin IX], QDs = quantum dots, MB = methylene blue.

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
