# Peer review of "Direct Electrochemical Detection of Glutamate, Acetylcholine, Choline, and Adenosine Using Non-Enzymatic Electrodes"

_sensors, 2019, doi:10.3390/s19030447_

Round 1
Reviewer 1 Report
The authors have reviewed the non-enzymatic electrochemical detection of non-electroactive neurotransmitters. The manuscript has some serious issues regarding the organization and scientific soundness. I suggest major revision and publishing in Sensors journal after significantly changing the format.
The manuscript lacks scientific discussion and reasoning and organization (usually seen in other review papers) and is mostly reporting some of the previously published results.
There is not any information regarding the concentration of the non-electroactive analytes in bodily fluids. The figures shown relate to mM concentration levels of the analytes which is too high to be real level in the body. More important is that the manuscript does not have any section dedicated to in-vivo studies on these neurotransmitter compounds detection.
All examples provided used highly alkaline solution as the detection medium, which can not be used for in-vivo detection. Isn't there any other report for neutral pH detection using these kind of nanomaterials??
The stability which has been claimed as one of the advantages for non-enzymatic methods are the same compared to the enzymatic sensors (table 1). Most of enzymatic sensors have stability in the order of 1 or 2 months while for the examples provided here the same stability has been reported. How you explain this?
Author Response
We sincerely appreciate the prompt feedback we received on our paper “Non-enzymatic Electrochemical Sensing of Nonelectroactive Neurotransmitters.” We have carefully considered the reviewer’s comments and have made the necessary changes to our submission. Below, you will find our responses to each reviewer comment.
Response to Reviewer #1 Comments
The authors have reviewed the non-enzymatic electrochemical detection of non-electroactive neurotransmitters. The manuscript has some serious issues regarding the organization and scientific soundness. I suggest major revision and publishing in Sensors journal after significantly changing the format.
Point 1: The manuscript lacks scientific discussion and reasoning and organization (usually seen in other review papers) and is mostly reporting some of the previously published results.
Response 1: Thanks for this crucial point. This paper focused on recent progress in non-enzymatic electrochemical detection of non-electroactive neurotransmitters. Few researches have been published in this area, and we tried to summarize and compare this kind of sensing with lots of published enzymatic sensing methods for the neurotransmitters in question. This review is primarily based upon available applicable studies but because the area is still relatively fresh, the list of applicable studies is short, there are some issues that have not been investigated. Accordingly, in the introduction section, we added some sentences to justify the issues you already mentioned (in red). In addition, we have added critical discussion at various locations in the manuscript to address the need for added scientific depth.
Point 2: There is not any information regarding the concentration of the non-electroactive analytes in bodily fluids. The figures shown relate to mM concentration levels of the analytes which is too high to be real level in the body. More important is that the manuscript does not have any section dedicated to in-vivo studies on these neurotransmitter compounds detection.
Response 2: As you correctly mentioned, there is no in vivo study in our paper because to the best of authors’ knowledge, there is no such information in all the reported researches in non-enzymatic detection of these kind of neurotransmitters. However, there are some in vivo studies using enzymatic methods. We added a section (section 5: In Vivo Applicability) to address this issue that the concentration of neurotransmitters in bodily fluids is much lower. The added section 5 also addresses the reviewer’s point 3 (neutral pH detection). In this section, those important in vivo studies, which used enzymatic methods, were also added. We also explained in the manuscript that the cyclic voltammograms in the figures were performed at non-physiological analyte concentrations for redox characterization.
Point 3: All examples provided used highly alkaline solution as the detection medium, which cannot be used for in-vivo detection. Isn't there any other report for neutral pH detection using these kind of nanomaterials??
Response 3: This is another important challenge of non-enzymatic detection of neurotransmitters that needs to be addressed. All the available experiments have been conducted in alkaline solution as mentioned in the added/expanded section 5. To contrast this limitation to enzymatic detection studies, we added reference to an enzymatic sensor tested in neutral solution.
Point 4: The stability which has been claimed as one of the advantages for non-enzymatic methods are the same compared to the enzymatic sensors (table 1). Most of enzymatic sensors have stability in the order of 1 or 2 months while for the examples provided here the same stability has been reported. How you explain this?
Response 4: In the table 1, we presented the best stability of enzymatic sensors and compared those with the available non-enzymatic sensors. In the section 4, where stability of the two methods has been compared, we added language to further explain that differences in testing parameters, number of experiments, and storage conditions make it difficult to draw stability comparisons. Also, we added a reference (https://doi.org/10.2116/analsci.30.213) showing that the lifetime of the immobilized enzymes and membrane materials, strength of immobilization or entrapment, can affect the long term stability.
Reviewer 2 Report
The authors of the manuscript titled “Non-enzymatic Electrochemical sensing of Non-electroactive Neurotransmitters” have presented a review article on the electrochemical detection of neurotransmitters such as glutamate, choline, acetylcholine and ATP directly at electrode surfaces (Ni electrodes) without any enzyme modifications.
We would recommend the following changes to the manuscript as a major revision:
1. One of the main advantages of using enzymatic sensors over non-enzymatic sensors is “selectivity”. On page 8, paragraph 4, section 4., While the authors have compared and contrasted the sensitivity of enzymatic sensors vs non-enzymatic sensors, they have not addressed the “selectivity” offered by enzymatic sensors. The authors must include a paragraph comparing and contrasting the “selectivity” of enzymatic vs non-enzymatic sensors.
2. Neurotransmitters such as glutamate, choline, acetylcholine and ATP are non-electroactive only at conventional electrodes. However, as covered in this review article, they undergo interesting redox behaviour at non-conventional electrodes such as Ni-based electrodes. My recommendation to the authors would be change the title of the manuscript to reflect this. Taking into account what’s been covered in this review article, a more suitable title for this article might be, for ex, “Direct electrochemical detection of neurotransmitters at non-enzymatic Ni-based electrodes”.
3. It is a well-written review article. However, there are minor errors/typos that need to be fixed. Page 4, paragraph 1, should be “indicative of its higher active surface area….”, page 6, paragraph 3, line 210, should be “Jamal et al., drop-casted a…”, page 8, paragraph 2, line 241, “aptameric sensors are non-enzymatic detectors using binding oligonucleotides that are highly specific to target analytes such as ATP.”
Recommendation: I would recommend the article for publication pending a major revision of the above comments.
Author Response
We sincerely appreciate the prompt feedback we received on our paper “Non-enzymatic Electrochemical Sensing of Nonelectroactive Neurotransmitters.” We have carefully considered the reviewer’s comments and have made the necessary changes to our submission. Below, you will find our responses to each reviewer comment.
Response to Reviewer #2 Comments
The authors of the manuscript titled “Non-enzymatic Electrochemical sensing of Non-electroactive Neurotransmitters” have presented a review article on the electrochemical detection of neurotransmitters such as glutamate, choline, acetylcholine and ATP directly at electrode surfaces (Ni electrodes) without any enzyme modifications.
We would recommend the following changes to the manuscript as a major revision:
Point 1: One of the main advantages of using enzymatic sensors over non-enzymatic sensors is “selectivity”. On page 8, paragraph 4, section 4., While the authors have compared and contrasted the sensitivity of enzymatic sensors vs non-enzymatic sensors, they have not addressed the “selectivity” offered by enzymatic sensors. The authors must include a paragraph comparing and contrasting the “selectivity” of enzymatic vs non-enzymatic sensors.
Response 1: Thanks for your comment. We briefly discussed the selectivity of the enzymatic and non-enzymatic sensors. However, we added more specific comparisons in this section (in red) to discuss selectivity of enzymatic sensors (section 4, paragraph 4).
Point 2: Neurotransmitters such as glutamate, choline, acetylcholine and ATP are non-electroactive only at conventional electrodes. However, as covered in this review article, they undergo interesting redox behaviour at non-conventional electrodes such as Ni-based electrodes. My recommendation to the authors would be change the title of the manuscript to reflect this. Taking into account what’s been covered in this review article, a more suitable title for this article might be, for ex, “Direct electrochemical detection of neurotransmitters at non-enzymatic Ni-based electrodes”.
Response 2: Thanks for your suggestion. In addition to Ni, different metals such as copper have been used in these non-enzymatic sensors. As you mentioned, a better title should convey this. So, we changed the title to "Direct Electrochemical Detection of Glutamate, Acetylcholine, Choline, and Adenosine Using Non-Enzymatic Electrodes". Clarification regarding the term “nonelectroactive” has also been added to the Introduction.
Point 3: It is a well-written review article. However, there are minor errors/typos that need to be fixed. Page 4, paragraph 1, should be “indicative of its higher active surface area….”, page 6, paragraph 3, line 210, should be “Jamal et al., drop-casted a…”, page 8, paragraph 2, line 241, “aptameric sensors are non-enzymatic detectors using binding oligonucleotides that are highly specific to target analytes such as ATP.”
Response 3: The grammatical errors have been fixed.
Round 2
Reviewer 1 Report
All my previous comments have been addressed in the current manuscript format and I strongly suggest its publication in Sensors journal.